# Automated Recognition and Measurement of Corrugated Pipes for Precast Box Girder Based on RGB-D Camera and Deep Learning

**DOI:** 10.3390/s25092641

**Published:** 2025-04-22

**Authors:** Jiongyi Zhu, Zixin Huang, Dejiang Wang, Panpan Liu, Haili Jiang, Xiaoqing Du

**Affiliations:** 1Department of Civil Engineering, School of Mechanics and Engineering Science, Shanghai University, Shanghai 200444, China; shuzhujy@shu.edu.cn (J.Z.); unuue@shu.edu.cn (Z.H.); djwang@shu.edu.cn (D.W.); 2Shanghai Highway and Bridge (Group) Co., Ltd., Shanghai 200433, China; liupanpan@srbg.com.cn (P.L.); jianghaili@srbg.com.cn (H.J.)

**Keywords:** corrugated pipes, recognition and measurement, semantic segmentation, deep learning, curve fitting

## Abstract

**Highlights:**

**What are the main findings?**
Corrugated pipes were identified using semantic segmentation techniques applied to reinforcement cage point clouds acquired by cost-effective RGB-D sensors.After the segmentation, registration, and fitting processes, an optimal configuration was proposed with maximum measurement errors < 6 mm.

**What is the implication of the main finding?**
It addresses the limitations of the automatic recognition of corrugated pipes caused by occlusion interference from rebars and precast molds.It replaces labor-intensive manual work in precast quality control, enhancing the efficiency and precision of the corrugated pipe recognition and measurement.

**Abstract:**

The accurate installation position of corrugated pipes is critical for ensuring the quality of prestressed concrete box girders. Given that these pipes can span up to 30 m and are deeply embedded within rebars, manual measurement is both labor-intensive and prone to errors. Meanwhile, automated recognition and measurement methods are hindered by high equipment costs and accuracy issues caused by rebar occlusion. To balance cost effectiveness and measurement accuracy, this paper proposes a method that utilizes an RGB-D camera and deep learning. Firstly, an optimal registration scheme is selected to generate complete point cloud data of pipes from segmented data captured by an RGB-D camera. Next, semantic segmentation is applied to extract the characteristic features of the pipes. Finally, the center points from cross-sectional slices are extracted and curve-fitting is performed to recognize and measure the pipes. A test was conducted in a simulated precast factory environment to validate the proposed method. The results show that under the optimal fitting scheme (BP neural network with circle fitting constraint), the average measurement errors for the three pipes are 2.2 mm, 1.4 mm, and 1.6 mm, with Maximum Errors of −5.8 mm, −4.2 mm, and −5.7 mm, respectively, meeting the standard requirements. The proposed method can accurately locate the pipes, offering a new technical pathway for the automated recognition and measurement of pipes in prefabricated construction.

## 1. Introduction

Prestressed concrete box girders are widely used in modern construction because of their superior torsion resistance, ease of mass production, and enhanced construction efficiency [1,2]. However, the installation position accuracy of a prestressed steel stand critically affects the structural performance of prestressed box girders [3,4]. The inaccurate installation position of pipes may lead to deviation of the prestressed steel stand, which may cause uneven stress distribution and component warping after the post-tensioning process, ultimately affecting the structural integrity and long-term durability of the component [5,6]. Current recognition and measurement methods face three key challenges: crisscrossing dense rebar, occlusion interference from rebars and precast molds, and narrow inspection spaces. Traditional measurement methods mainly rely on manual operation, using a tape measure for sampling inspections. However, this approach is labor-intensive, subject to operator bias, and inefficient [7,8]. These limitations underscore the imperative for a precise installation position to ensure quality assurance in prestressed concrete components.

In recent years, non-destructive testing technologies and specialized detection instruments have been increasingly applied to locate corrugated pipes in prestressed structures. Non-destructive testing technology has been gradually applied to locate corrugated pipes, primarily focusing on the position of embedded pipes after concrete pouring. The current methodologies fall into three categories: (1) the impact echo method [9]; (2) the ground penetrating radar method [10,11,12]; and (3) the infrared ray method. The current research on pre-pouring pipe inspection predominantly concentrates on specialized instrument development rather than automated recognition method solutions. For instance, Guo Y. [13] designed a three-dimensional localization instrument for corrugated pipes capable of measuring vertical, horizontal, and longitudinal coordinates. Similarly, Guo Q. et al. [14] integrated a magnetic grid measurement system with a phase laser range-finder instrument, enabling real-time data transmission via mobile applications and remote monitoring through a web site. In summary, existing research has primarily concentrated on non-destructive testing after concrete pouring. While these post-pouring inspections provide quality verification, pre-pouring pipe inspection offers greater engineering value by enabling corrective adjustments during prestressed steel stand installation. However, research in this domain remains limited, with most efforts on developing specialized instruments, which still rely heavily on manual operation and manual data reading. For corrugated pipe measurement tasks spanning up to thirty meters, existing solutions in practical engineering have failed to effectively address critical challenges such as low detection efficiency, high manual labor costs, and inadequate detection accuracy. Furthermore, the detection process, which requires workers to stand on reinforcement mesh to inspect deeply embedded corrugated pipes, poses substantial safety risks to the workers’ occupational health and working conditions. These limitations highlight the urgent need for intelligent, automation-driven recognition methods for pre-pouring pipes in precast concrete factories. Consequently, it is imperative to develop an automated corrugated pipe measurement approach that enhances detection precision and efficiency while reducing substantial safety risks during the inspection process.

With the rapid development of artificial intelligence (AI) technology, machine vision and deep learning technology have been increasingly integrated into the field of engineering applications, aiming to enhance construction labor productivity and efficiency [15,16], such as quality inspection [17,18,19,20,21], 3D model reconstruction [22,23,24,25,26], schedule monitoring [27,28,29,30,31,32], and safety management [33,34]. In engineering quality inspection, tasks such as concrete crack detection [35], component dimensional measurement [36], and defect detection [37] are used. In the domain of pre-pouring corrugated pipe detection for precast components, Shu et al. [38] proposed an innovative method for pipe point cloud extraction using 3D laser scanning sensors (FARO^®^ Focus S350 from FARO Technologies, Inc., in Lake Mary, FL, USA) integrating radius nearest-neighbor covariance characteristics with clustering algorithms. However, vision-based technologies for pre-pouring corrugated pipes remain remarkably limited with few relevant studies. The lack of automated visual detection approaches represents a significant technological constraint in engineering applications, calling for developing additional innovative and cost-effective solutions for corrugated pipe measurement with the aim of advancing practical implementation and efficiency in precast factories.

Point cloud data, as one of the most common forms of high-dimensional data, have been widely used for spatial perception and geometric analysis owing to the capability of these data to rapidly and accurately capture information from object surfaces [39,40,41]. Three-dimensional laser scanners [42] and RGB-D cameras [43] are widely used among the different types of acquisition devices. Three-dimensional laser scanners are notable for their ability to acquire large-scale three-dimensional spatial data with high precision and excellent stability [44]. Kim et al. [45] used a laser scanner to achieve the automatic dimensional measurement of simple geometric shape components such as precast concrete slabs, in terms of length, width, and verticality. Li et al. [46] used laser scanners to inspect the full-surface dimensional quality of planar precast components without point cloud registration. On the other hand, RGB-D cameras offer advantages in portability, flexible application design, low cost, and short-range accuracy [47]. Bellés et al. [48] proposed an automatic 3D modeling system for sewer manholes using a Kinect sensor, while Rebolj et al. [49] used a Kinect 2 camera (Microsoft Corp., in Redmond, WA, USA) to realize the automatic detection of building elements and construction progress monitoring.

Although the measurement accuracy of RGB-D cameras is lower than that of laser scanners, they excel at short-range detection. In contrast, 3D laser scanners are ideal for capturing detailed information over long distances; however, they are more susceptible to noise and occlusion, especially when scanning complex areas such as the interiors of rebar structures and precast molds, which can reduce inspection accuracy. Moreover, laser scanners typically lack portability, limiting their practical applications. While industrial-grade portable laser scanners are beginning to be introduced into automated assembly systems, their high costs and economic challenges make them difficult to implement in precast factories. Overall, RGB-D cameras are better suited for acquiring data on corrugated pipes due to their cost effectiveness, compactness, and real-time performance. However, since RGB-D cameras do not capture comprehensive site-scale information, point cloud registration must be performed during post-processing to accurately reconstruct the full geometry of corrugated pipes. Current research primarily focuses on single-frame point cloud registration for small-scale, simple scenes, and studies addressing the challenges in more complex environments, such as those encountered in precast factories, remain limited.

Several methods based on point cloud data have been proposed for inspecting structural components [50,51,52]. However, research on pipe inspection remains limited. Deep learning can directly process 3D point cloud data through multi-layer neural networks, enhancing both extraction accuracy and efficiency. Qi et al. [53] proposed a PointNet network to directly process 3D laser point cloud data while maximally preserving the spatial features of the point cloud. Subsequently, Deng [54] applied the PointNet network for semantic segmentation in railway tracks, successfully extracting railway track lines. Hu et al. [55] proposed the RandLA-Net for processing large-scale point clouds. Some researchers have applied the RandLA-Net network for applications such as the longitudinal deformation detection of tunnels [56], highway guardrail trajectory reconstruction [57], and the segmentation and localization of storm trains in urban drainage systems [58]. While existing deep learning approaches exhibit robust performance and feasible ideas in pipe-shaped detection tasks, significant research gaps persist regarding the segmentation of corrugated pipes. The critical challenge stems from the significant occlusion interference caused by rebars and precast molds.

In summary, this paper introduces an automated method for recognizing and measuring corrugated pipes in precast box girder components using a cost-effective RGB-D camera. First, the point cloud data captured by the camera are merged using a point cloud registration algorithm. Following data preprocessing, a semantic segmentation network is employed to extract the complete point cloud of the pipes. By integrating deep learning techniques with the pipes’ geometric features, center point and curve fitting methods are applied to achieve precise measurements. The experimental validations demonstrate that this approach delivers efficient and accurate inspections of corrugated pipes while significantly enhancing automation and overall process efficiency. This work offers new insights and technical avenues for the automated recognition and measurement of corrugated pipes in precast environments.

The key contributions of this study are summarized as follows:(1)This paper develops an innovative integrated method for precise corrugated pipe recognition and measurement, combining registration algorithms, semantic segmentation, and fitting techniques, successfully meeting engineering standard requirements.(2)This paper comprehensively evaluates the applicability of different registration algorithms and compares the performance between traditional and neural network approaches in corrugated pipe curve fitting, providing valuable references for future pipe measurement research.(3)The proposed methodology addresses manual detection limitations by improving efficiency, reducing subjectivity, and enhancing operational safety in prestressed corrugated pipe measurement.(4)This paper proposes a cost-effective automated measurement solution using RGB-D sensors, offering an economically viable paradigm for advancing the practical implementation of automated pipe measurement technologies.

## 2. Methodology

Figure 1 shows the framework of the proposed corrugated pipe measurement method. The process begins with the acquisition of point cloud data using an RGB-D camera, followed by the construction of a dataset. Since point cloud coordinates obtained by RGB-D cameras are referenced from individual camera shooting positions, point cloud registration is essential to transform all data into a unified coordinate system. To balance computational efficiency and accuracy, the optimal fine registration algorithm was selected. By integrating coarse and fine registration methods, the complete point cloud data of pipes was generated. Then, considering that corrugated pipes are deeply embedded within rebars and precast molds, this study addresses the critical issue of severe point cloud data missing and interference. To overcome these limitations, a large-scale semantic segmentation algorithm with robust capabilities in handling point cloud deficiencies was used to extract the point cloud data of the corrugated pipes from the raw point cloud. To reduce computational burden and enhance computational efficiency, the segmented pipe data were sliced along the length of the girder for further analysis. To simplify the prediction process, the center point of each slice was extracted and curve fitting was conducted, effectively streamlining the pipe measurement task. Taking into account the cross-sectional characteristics of corrugated pipes and slice symmetry, the center point of each slice was determined by using the axis align bounding box method and the circle fitting constraint method. Finally, by leveraging the center point information, we conducted a comprehensive comparative analysis between traditional curve fitting methods based on pipe geometric features and advanced neural network-based fitting approaches, systematically evaluating their respective performance and applicability. The backpropagation (BP) neural network and polynomial curve fitting were used to fit the centerline of the pipes. The height coordinates of these points were calculated to inspect the accuracy of pipe measurement. Comparing the final fitted values with actual measurements and evaluating them against metrics, we selected the optimal pipe fitting scheme.

## 3. Data Preparation and Registration

### 3.1. Data Acquisition and Dataset Construction

In this paper, a Zivid One+ L camera (Zivid Technologies, in Oslo, Norway) was used for point cloud data acquisition. It relies on structured light projection to illuminate the target and computes depth information by analyzing the deformation of reflected light patterns. The camera can output both RGB images and dense point cloud data, each containing XYZ (spatial coordinates), RGB (color), and SNR (signal-to-noise ratio) information. The main specifications and parameters of the camera are described in Table 1.

Corrugated pipes usually occupy significant spatial volumes in precast factories, as shown in Figure 2. Therefore, to efficiently capture the pipe data using an RGB-D camera, a segmental capturing strategy was adopted. A precast segmented box girder model was designed and fabricated to replicate the production environment of box girders in a Shanghai precast factory.

Figure 3 shows the 3D design of the segmental model, while Figure 4 shows the actual scene. The model incorporated the following items: (1) rebar arrangement—rebars were tied in a diagonal pattern from top to bottom and left to right, closely mimicking on-site conditions; (2) adjustable pipes—the pipes could be adjusted to simulate their profiles in different segmental girders; (3) concrete support—precast concrete pads were installed in designated areas to replicate structural constraints. Data acquisition was carried out on both the front and back of the model, respectively.

### 3.2. Point Cloud Registration

#### 3.2.1. Algorithm Introduction

The point cloud data captured by the RGB-D camera are inherently relative to the camera’s coordinate system at the time of capture. Registration is therefore required to unify all data into a global coordinate system. This process is divided into two stages: coarse registration and fine registration.

In this paper, the Random Sample Consensus (RANSAC) algorithm combined with Fast Point Feature Histograms (FPFHs) was used as the coarse registration algorithm for initial alignment. RANSAC estimates the transformation matrix by iteratively identifying optimal correspondences between point clouds, while FPFHs extract local geometric features (e.g., surface curvature and normal vectors) to enhance matching robustness. Their integration enables efficient estimation of the initial transformation matrix.

Then, three different fine registration methods were compared for precise alignment. In the fine registration stage, the traditional Iterative Closest Point (ICP) algorithm is widely used because of its simplicity and effectiveness. However, this method only considers the Euclidean distance between point pairs, neglecting the structural information of the point cloud. In order to improve the configuration accuracy, the Point-to-Plane ICP algorithm has been proposed. Point-to-Plane ICP minimizes the distance from points to planes when calculating the distance between the source point cloud and the target point cloud. Although this method improves accuracy, it significantly increases computational complexity. The GICP (Generalized ICP) algorithm extends ICP by integrating normal vector information, achieving higher accuracy at the cost of elevated computational load. In addition, this paper also explores a color-assisted registration method based on color features to improve robustness. Color information enhances registration precision by constraining point-to-point alignment along the tangent through multi-conditional error optimization. However, this approach is susceptible to environmental noise and lighting variations. The complexity arises from simultaneously processing spatial geometric features and color information, which increases computational overhead and convergence time.

#### 3.2.2. Evaluation Metrics

Two widely used evaluation metrics, fitness and Root Mean Square Error of Inliers (RMSE_Inlier_), were used to evaluate registration accuracy. The source and target point clouds with overlapping regions were registered, and the results are quantified in Table 2, with visualizations provided in Figure 4.

(1)Fitness

Fitness represents the alignment quality between registered point clouds, defined as the ratio of matched inliers in the source point cloud to the total number of points in the target point cloud. Higher fitness values indicate better registration performance.

(2)Root Mean Square Error (RMSE_Inlier_)

RMSE_Inlier_ represents the geometric consistency of aligned inlier pairs by calculating the Root Mean Square Error of their spatial deviations. Lower RMSE_Inlier_ values correspond to higher registration precision and lower deviations.

As shown in Table 2, Point-to-Plane ICP and Generalized ICP (GICP) achieve comparable results, both demonstrating high registration quality (fitness ≈ 0.37, RMSE_Inlier_ < 1.5 mm). In contrast, color-assisted registration exhibits significantly lower fitness (3.19 × 10^−6^) and prolonged registration time (>60 s), rendering it unsuitable for pipe registration.

Figure 5 illustrates a visual comparison of the registration results. The Point-to-Plane ICP method failed to align the left-side regions, while both Point-to-Plane ICP and color-assisted methods showed noticeable misalignment between pipes and rebars, with offset gaps. GICP achieved optimal alignment with minimal offset gaps. Although GICP required approximately three times the registration time of Point-to-Plane ICP (0.188 s vs. 0.056 s), its total processing time remained practical for construction applications. Based on registration results and evaluation metrics, GICP was selected as the primary fine registration method.

Figure 6 illustrates the registration results of 70 segmented corrugated pipes and provides a local magnified view of the registration result before semantic segmentation. The results demonstrate high registration accuracy with minimal noise points. The three corrugated pipes retained their slender geometric features. However, local missing areas still existed due to occlusion interference, capturing angle and height limitations. Despite these challenges, the overall curve of the pipes maintains good integrity.

### 3.3. Point Cloud Preprocessing

To improve data quality, reduce data volume, and enhance processing efficiency, preprocessing was performed on the registered point cloud data. The preprocessing steps include point cloud thinning, manual segmentation, and annotation, which were implemented in Cloud Compare software v2.13.2.

First, the registered point cloud data were divided into 0.5 m intervals along the girder length. Then, spatial down-sampling with a spacing of 0.0035 m was applied to reduce the point cloud data of each segment to around 500,000. Next, based on a visual estimate, the divided data of the precast box girder were classified into corrugated pipes and non-pipes. Subsequently, the labels were manually annotated and merged into a unified TXT file, preserving RGB (color) and XYZ (spatial coordinates) information.

During the dataset construction process, 120 groups of data were randomly selected from the preprocessed point clouds as the training set, and 10 groups from the same corrugated pipes were selected as the test set. To ensure the independence and reliability of model evaluation, the training set and test set were strictly separate, with no overlap between them.

## 4. Recognition and Measurement of Corrugated Pipes

### 4.1. Semantic Segmentation of Pipes

Currently, the predominant point cloud semantic segmentation algorithms primarily include PointNet, PointNet++, and RandLA-Net. PointNet and PointNet++ require preprocessing input data into 1 m × 1 m blocks, making them more suitable for small-scale point clouds and tasks demanding high semantic segmentation detail. However, prestressed corrugated pipes, characterized as slender cylindrical structures with lengths potentially reaching thirty to forty meters in prestressed box girders, require algorithms oriented towards large-scale scene segmentation.

Moreover, the point cloud data of prestressed corrugated pipes embedded within rebars and precast molds are often incomplete. While PointNet and PointNet++ are sensitive to missing point cloud data, RandLA-Net offers solutions through its random sampling methodology, effectively mitigating point cloud deficiency issues. Additionally, its local attention mechanism can substantially suppress noise points, maintaining high segmentation accuracy in complex environments.

Corrugated pipes are slender and can reach 30–40 m in precast box girders. To better preserve the complete geometric features of the pipes and meet a factory’s demand for efficient real-time detection, this paper selected RandLA-Net, a lightweight network optimized for large-scale 3D point clouds, as the semantic segmentation model.

RandLA-Net uses an encoder–decoder architecture to efficiently process point clouds of up to one million points. In the encoder, random sampling reduces the points (e.g., sampling 10⁵ points from 10⁶ in 0.004 s), while a Local Feature Aggregation (LFA) module, comprising Local Spatial Encoding, Attentive Pooling, and Dilated Residual Blocks, mitigates feature loss and captures complex local structures. The decoder up-samples using k-nearest-neighbor interpolation, aggregates multi-scale features via skip connections, and applies a shared MLP to reduce dimensions and restore density, with final predictions generated through fully connected layers and dropout regularization. Figure 7 shows the network, with N indicating the number of points and dashed lines representing skip connections.

### 4.2. Semantic Segmentation Results and Analysis

The computer operating system used for training was Ubuntu 18.04 (64 bit). The graphics card used for training was an NVIDIA GeForce RTX 3090 (NVIDIA Corp., in Santa Clara, CA, USA) with 24 GB VRAM. The central processing unit (CPU) was Intel^®^ CoreTM i7-7800X(Intel corp., in Santa Clara, CA, USA), equipped with 16 GB RAM. The software environment was configured with Python 3.6, TensorFlow2.6-GPU, CUDA 11.4, and cuDNN 8.2.2.26.

The model was trained for 100 epochs with a batch size of 6, an initial learning rate of 0.01, and a 5% learning rate decay per epoch. The model processed 40,960 points per iteration. The k-nearest-neighbor parameter was set to k = 16, the loss function was the cross-entropy loss, and the optimizer was the Adam optimizer.

To validate the effectiveness of the proposed method, this paper used four evaluation metrics: Mean Accuracy (mAcc), Intersection over Union (IoU), Mean Intersection over Union (mIoU), and Overall Accuracy (OA), where mAcc represents the average proportion of correctly classified points across all categories, IoU represents the ratio of the intersection to the union between the ground truth and predicted values for each category, mIoU represents the average IoU across all categories, and OA represents the proportion of correctly predicted points relative to the total number of points. The calculation formulas for each metric are as follows:(1)$$mAcc=\frac{1}{k}\sum_{i=1}^{k}\frac{\;P_{ii}}{\sum_{j=1}^{k}\;\;P_{ij}}$$(2)$$OA=\frac{\sum_{i=1}^{k}\;P_{ii}}{\sum_{i=1}^{k}\;\sum_{j=1}^{k}\;P_{ij}}$$(3)$$IoU=\frac{P_{ii}}{\sum_{j=1}^{k}\;P_{ij}+\sum_{j=1}^{k}P_{ji}-P_{ii}}$$(4)$$mIoU=\frac{1}{k}\sum_{i=1}^{k}\;\frac{P_{ii}}{\sum_{j=1}^{k}\;P_{ij}+\sum_{j=1}^{k}P_{ji}-P_{ii}}$$

In these formulas, *k* represents the total number of semantic categories, P*_ii_* refers to true positive (the number of points correctly classified as category *i*), and P*_ij_* and P*_ji_* represent the false negatives and false positives (the number of misclassified points).

The prediction results of the model on the provided dataset are shown in Table 3, and the localized result of semantic segmentation is illustrated in Figure 8. As shown in Table 3, the proposed model achieves superior segmentation performance, producing segmented point clouds with minimal noise and preserving the essential curve and contour features of the original data. Notably, the IoU for pipes is 99.92%, while the IoU for non-pipes is 99.96%; these high scores stem from the stable indoor environment, which yielded high-quality training data.

In conclusion, based on the point cloud data of the box girder acquired by the RGB-D camera, the RandLA-Net network effectively extracts pipe point cloud data. These results validate the feasibility of the proposed model for the location of corrugated pipes in precast box girders.

### 4.3. Center Point Extraction and Curve Fitting Methods

To enhance data processing efficiency, the point cloud data of the pipes were first segmented into slices along the girders’ length. The center point of each slice was then extracted and utilized in curve fitting to determine the pipes’ centerline. A comparative study was conducted on various center point extraction and curve fitting methods, with the objective of identifying the optimal techniques for accurately locating the pipes.

Considering that corrugated pipes are deeply embedded within rebars and precast molds, certain regions located in visual blind spots are prone to the loss of point cloud data. The cross-sectional profile exhibits a short-arc characteristic. To address this challenge, we employed an iterative approach, fitting the short arc to a circle and utilizing the minimization of the circular radius as a constraint to determine the circle’s center. Concurrently, pipe slices approximate symmetry around their bounding box center, thus motivating our selection of the bounding box center method as an alternative positioning strategy.


**Center Point Extraction Methods**


Two center point extraction methods were employed: the axis aligned bounding box method and the circle fitting constraint method. The axis aligned bounding box method encloses the set of points within the smallest possible rectangular region whose edges are parallel to the coordinate axes. In this approach, the sliced pipe point cloud was first projected into a local coordinate system. Next, the extreme values along each axis were calculated to construct the bounding box. Finally, the center of the box was obtained by averaging these minimum and maximum values, providing an accurate estimation of the pipe’s central point in that slice. The centroid of the bounding box was regarded as the pipe center point, donated as P*_i_* (*i* = 1, 2, 3, 4, …, *n*), serving as the center point of a slice, and the Z-coordinate of each point corresponds to the vertical positioning height of the pipe, as shown in Figure 9.

The prestressed corrugated pipe features a circular cross-section with a known nominal radius. To efficiently locate its central axis, the circle fitting constraint method leverages this geometric by optimizing a least-squares solution across the sliced point cloud. The algorithm operates as follows: for each planar slice projected onto the XOZ-coordinate system, it calculated the Euclidean distance between every sampled point *K_i_*(*x_i_,y_i_*) and a candidate circle center. By minimizing the sum of squared residuals between these distances and the predefined radius, the method converged to an optimal center position that either intersects or closely approximates the majority of data points. Mathematically, this constrained optimization ensured the fitted circle adhered to the fixed radius while maximizing spatial agreement with the observed point distribution. The resulting center coordinates *P_i_* (*i* = 1, 2, 3, 4, …, *n*) correspond to sequential cross-sectional centroids along the pipe’s length. Crucially, the Z-coordinate of each directly encoded the vertical positioning height of the pipe, as visualized in Figure 9.

2.
**Curve Fitting Methods**


In curve fitting methodologies, the regular geometric features of corrugated pipes can be approximated as a quadratic parabola. This study comprehensively investigates different curve fitting approaches, systematically comparing traditional geometry-based curve fitting methods with neural network curve fitting techniques. While traditional methods rely on rigorous mathematical models of geometric features, neural network approaches offer a more flexible nonlinear curve fitting pathway capable of effectively capturing complex geometric variation characteristics.

The centroid point set obtained in the preceding section undergoes parametric curve fitting. Conventional approaches establish mathematical relationships between data points through predefined functional models and optimized parameters. Common methods include least squares approximation, polynomial regression, and spline interpolation. Given that the geometric profile of corrugated pipes approximates a quadratic parabola, polynomial fitting is selected as one of the fitting strategies where the polynomial degree is constrained between 3 and 7 to balance precision and overfitting risks.

For enhanced nonlinear modeling capability and automated feature learning, neural networks are concurrently implemented. The backpropagation (BP) neural network operates without presumptive functional forms, exhibiting superior adaptability to high-dimensional data and complex nonlinear relationships. The network takes centroid coordinates as the input and outputs predicted height values, employing a 128 × 64 × 1 layered architecture with tanh and ReLU activation functions in hidden layers and linear activation in the output layer. Training utilizes the Adam optimizer to minimize Mean Squared Error (MSE) loss, and the learning rate was 0.001.

## 5. Experimental Results and Analysis

According to the JTG F80/1-2017 [59] standard, the allowable vertical installation deviation for pipe detection is ±10 mm. To thoroughly evaluate the performance of the proposed pipe measurement method on an entire precast box girder, a simulated installation environment replicating actual factory conditions was created for both training and testing. In this environment, three corrugated pipes—labeled A, B, and C from top to bottom—were installed along the girder. Due to the non-standardized geometries of the spliced prestressed pipes, which include variable linear, curved, and horizontal segments, 12 sampling points were established at 3 m intervals along each pipe for experimental validation, with true heights measured using a tape measure. The fitting methods for the three pipes were then evaluated using the Mean Absolute Error (MAE), Root Mean Square Error (RMSE), and Maximum Error (Max_error_) metrics to quantify the deviations between the fitted values and the ground truth. Table 4 presents the performance metrics for each fitting scheme.

The quantitative analysis showed that under the optimal fitting scheme (circle fitting constraint method with BP neural network), the MAES for the three pipes were 2.2 mm, 2.4 mm, and 1.6 mm, respectively, and the Maximum Detection Errors were −5.8 mm, −4.2 mm, and −5.7 mm, respectively, all of which met the requirements of the standard. However, in most of the fitting results, the Maximum Error occurred in the regions between points 4–5 and points 8–10, corresponding to the pipes transitioning from bending to restoring the horizontal position and from the horizontal position to restoring the bent state. These areas pose challenges for composite curve fitting. In addition, the regions have severe occlusions caused by rebars and precast molds, which lead to missing or noisy point cloud data, further increasing the fitting error.

Figure 10, Figure 11 and Figure 12 showcase the results of various fitting schemes applied to the three pipes. These figures provide a clear comparison of the performance of the center point extraction methods (i.e., the circle fitting constraint method and the axis aligned bounding box method) and curve fitting approaches (i.e., BP neural network and polynomial curve fitting). The results highlight that the BP neural network consistently outperformed polynomial curve fitting, particularly in regions with curved segments and near the pipe ends.

Notably, for Pipe C, located at the bottom of the girder, the prediction curve derived from the polynomial fitting method exhibited significant bias when compared to the BP neural network curve, failing to meet the standard requirements. This discrepancy can be attributed to the severe occlusion caused by rebars and precast molds at the girder’s bottom. The occlusion significantly impeded point cloud acquisition, resulting in sparse and inconsistent data. Consequently, as shown in Figure 12, the center points extracted using both the circle fitting constraint method and the axis aligned bounding box method were highly scattered. The dispersed center points impaired the polynomial curve fitting’s ability to produce an accurate prediction, leading to substantial error. In contrast, the BP neural network fitting demonstrated superior robustness, effectively handling the dispersed center points and maintaining the fitting bias within the permissible error range. This ensured ideal fitting performance, even under challenging conditions.

## 6. Discussion and Conclusions

This paper constructed a dataset through environmental simulation of precast factory conditions. The point cloud data were captured using an RGB-D camera, and multi-registration algorithms generated the complete data. The RandLA-Net network was applied for the semantic segmentation of pipes. Through center point extraction and curve fitting methods, the recognition and measurement of the pipes were achieved. The key findings of this paper are summarized as follows:Under the proposed optimal fitting method (BP neural network combined with the circle fitting constraint method), the average detection errors for the three pipes were 2.2 mm, 1.4 mm, and 1.6 mm, respectively, with maximum detection errors of −5.8 mm, −4.2 mm, and −5.7 mm, all of which meet the requirements of construction standards.The proposed point cloud registration method integrates both coarse and fine registration stages to achieve accurate alignment. During the coarse registration stage, the RANSAC algorithm combined with Fast Point Feature Histograms (FPFHs) was employed for initial alignment by iteratively estimating optimal feature correspondences. In the fine registration stage, three methods, Point-to-Plane ICP, Generalized ICP (GICP), and color-assisted point cloud registration, were evaluated and compared. Based on key metrics, including alignment accuracy, the RMSE of inlier pairs, and computation time, GICP was identified as the most suitable method for the fine registration of point clouds in precast segmental box girders.BP neural networks and polynomial curve fitting were employed to model the centerlines of the pipes. The BP neural network demonstrated significantly greater robustness, effectively managing the irregular and dispersed center points in the data over traditional polynomial curve fitting. It consistently maintained fitting biases within the permissible error range, achieving reliable and accurate performance even under challenging conditions.

## 7. Limitation and Future Research

The results of this paper on the recognition and measurement of corrugated pipes using RGB-D cameras were found via experiments conducted in controlled indoor environments for bridge girder prefabrication factories. However, to achieve broader adoption and practical application, this method must be adapted to a wider range of construction scenarios. Future research should prioritize the development of high-precision methods capable of addressing the challenges posed by complex environmental conditions, including variations in weather, lighting, and equipment limitations. Additionally, tackling issues such as occlusion caused by rebars and precast molds, as well as managing incomplete or missing data, will be critical for enhancing the algorithm’s robustness. These advancements will be key to ensuring the method’s reliability and effectiveness in real-world construction environments.

## Figures and Tables

**Figure 1 sensors-25-02641-f001:**
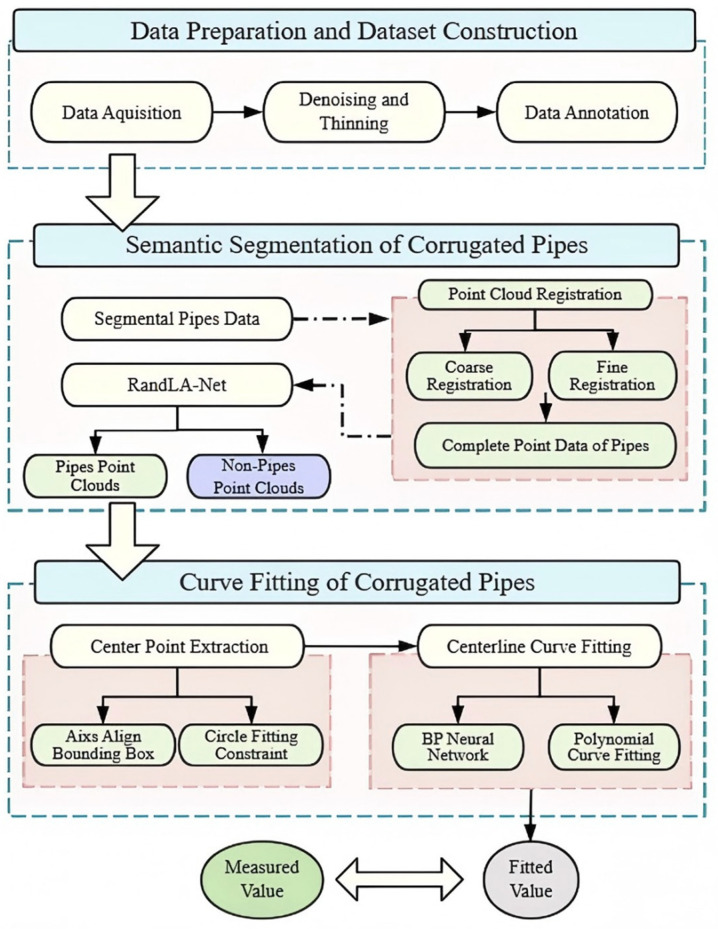
Recognition and measurement framework for corrugated pipes.

**Figure 2 sensors-25-02641-f002:**
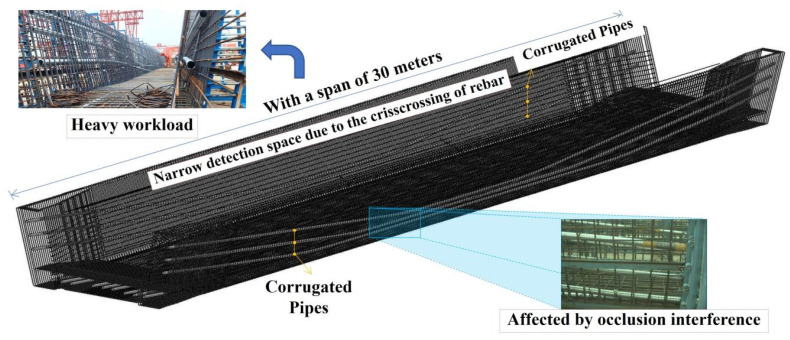
Scene diagram of corrugated pipes.

**Figure 3 sensors-25-02641-f003:**
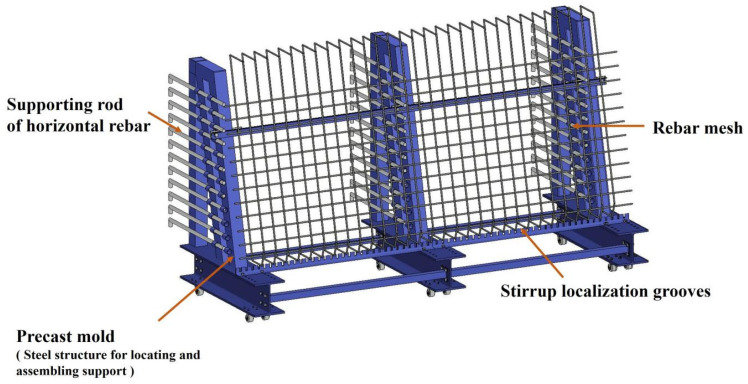
Three-dimensional design of segmental model of precast box girder.

**Figure 4 sensors-25-02641-f004:**
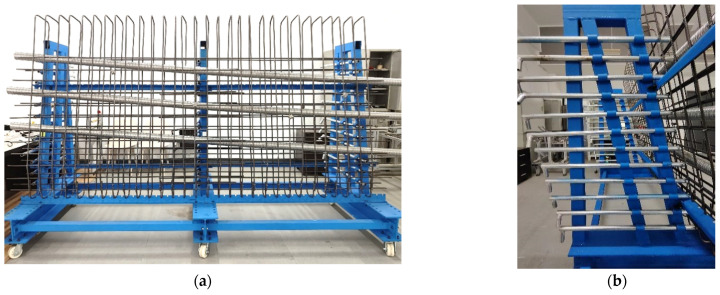
Actual view of segmental model of precast box girder: (**a**) front view; (**b**) side view.

**Figure 5 sensors-25-02641-f005:**
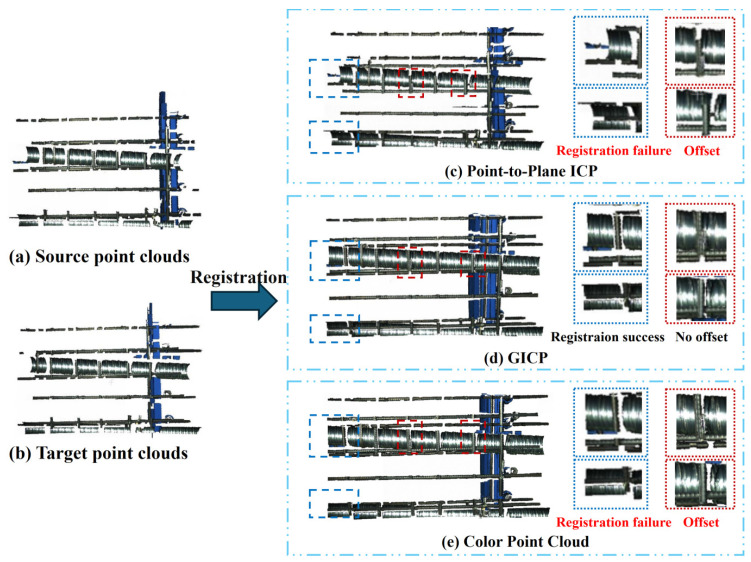
Registration results of two groups of point clouds with different algorithms.

**Figure 6 sensors-25-02641-f006:**
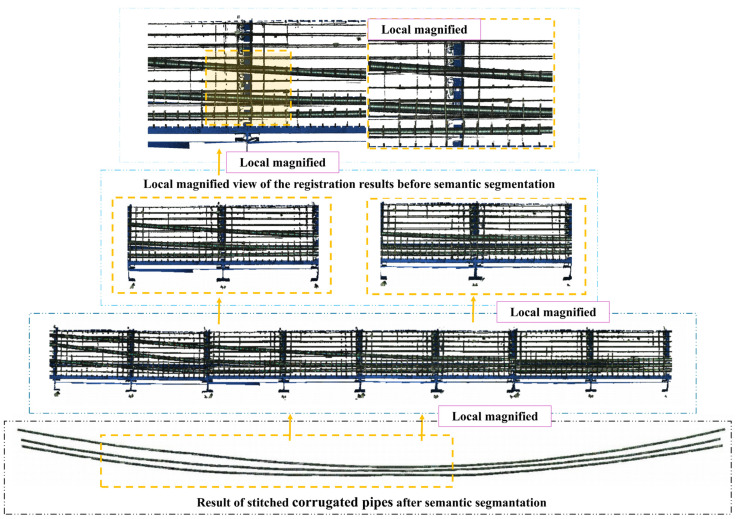
Display of the pipes after registration.

**Figure 7 sensors-25-02641-f007:**
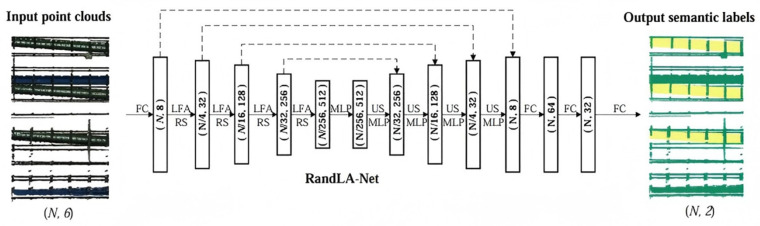
Network architecture of RandLA-Net.

**Figure 8 sensors-25-02641-f008:**
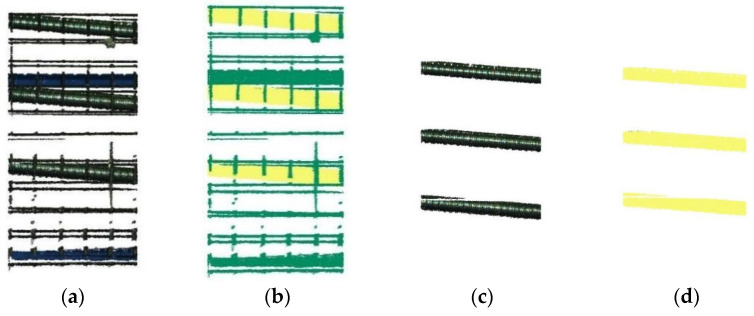
Semantic segmentation result: (**a**) original point cloud; (**b**) predicted point cloud; (**c**) annotated bellows; (**d**) predicted bellows.

**Figure 9 sensors-25-02641-f009:**
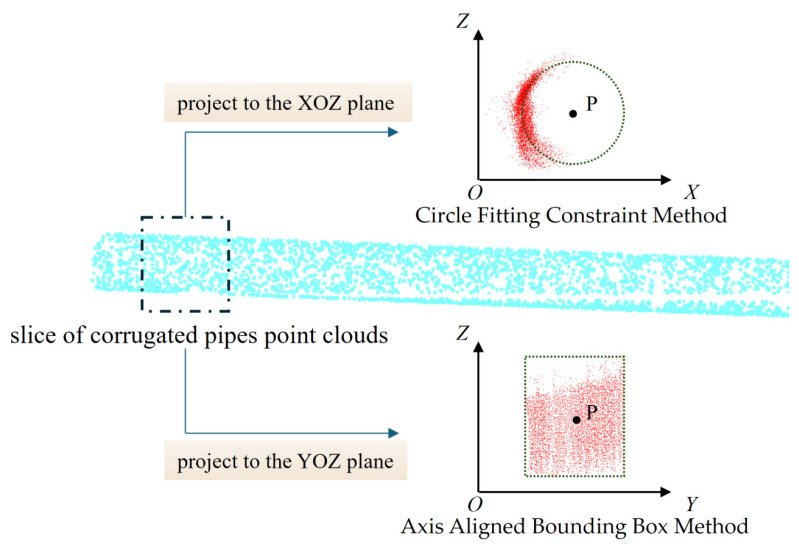
Examples of the two extraction methods. P represents the position of the center point, blue is the three-dimensional bellows point cloud, and red is the point cloud after projection.

**Figure 10 sensors-25-02641-f010:**
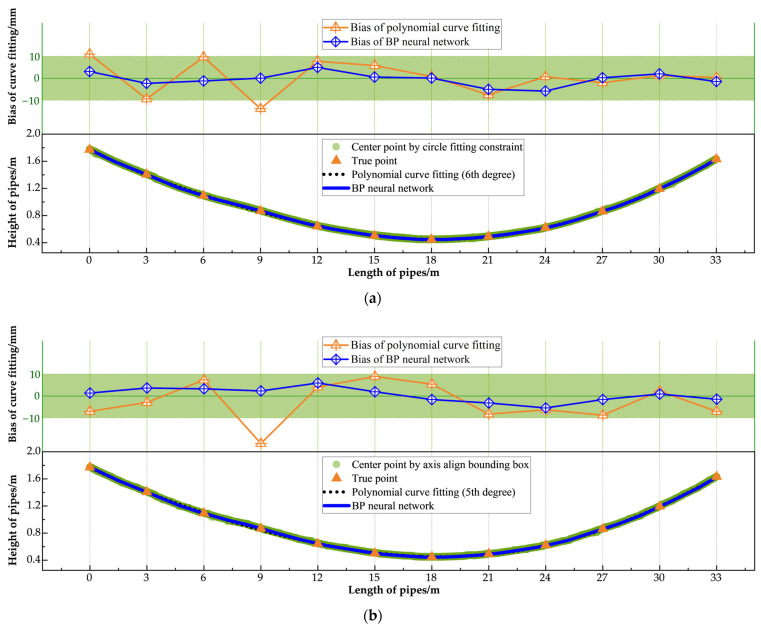
Fitting result of No. A pipe: (**a**) center point extracted by circle fitting constraint method; (**b**) center point extracted by axis align bounding box method.

**Figure 11 sensors-25-02641-f011:**
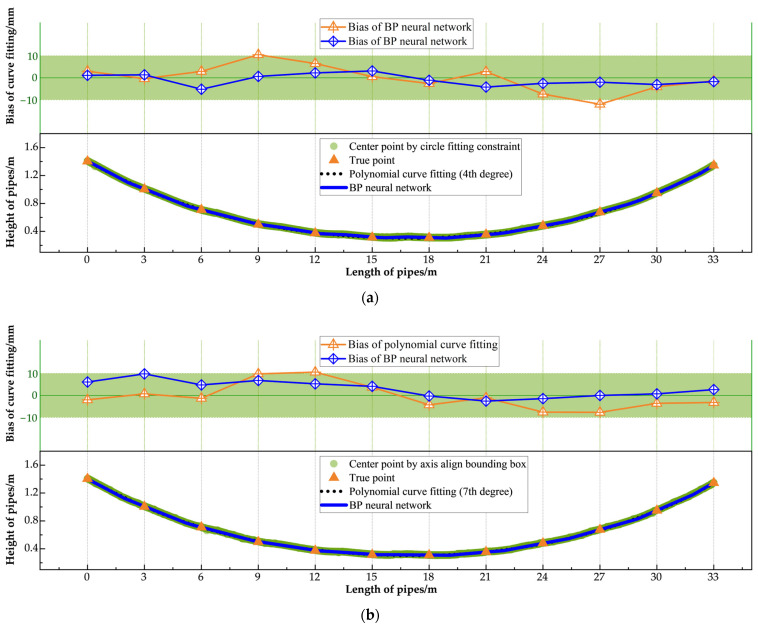
Fitting result of No. B pipe: (**a**) center point extracted by circle fitting constraint method; (**b**) center point extracted by axis align bounding box method.

**Figure 12 sensors-25-02641-f012:**
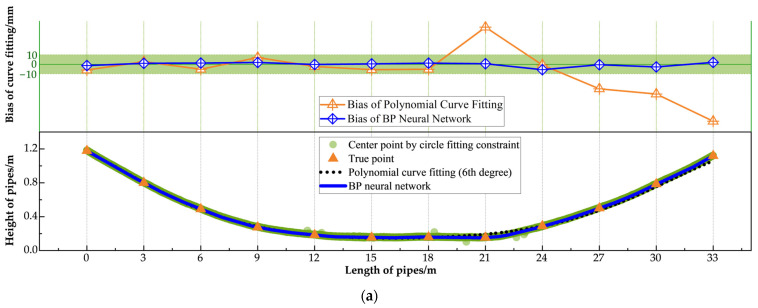
Fitting result of No. C pipe: (**a**) center point extracted by circle fitting constraint method; (**b**) center point extracted by axis align bounding box method.

**Table 1 sensors-25-02641-t001:** Specifications of the Zivid One+ L camera.

Model	Detection Range (m)	Field of View (mm)	Color Resolution	Spatial Resolution (mm)
Zivid One+ L 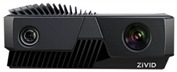	1.2–3	2069 × 1310@3000	1920 × 1200	1.11@300

**Table 2 sensors-25-02641-t002:** Results of different fine registration algorithms.

Algorithms	Fitness	RMSE_Inliner_ (mm)	Registration Time (s)
Point-to-Plane ICP	0.378	1.47	0.056
GICP	0.374	1.46	0.188
Color assisted	3.19 × 10^−6^	7.52	72.733

**Table 3 sensors-25-02641-t003:** Prediction results of the testing set.

Model Name	OA (%)	mAcc(%)	mIoU (%)	IoU_bellows_ (%)	IoU_nonbellows_ (%)
Model	99.97	99.96	99.94	99.92	99.96

**Table 4 sensors-25-02641-t004:** Fitting results for each scheme.

Fitting Scheme	No. A Bellow				No. B Bellow				No. C Bellow			
	MAE	RMSE	Max_error_ (mm)	P *	MAE	RMSE	Max_error_ (mm)	P *	MAE	RMSE	Max_error_ (mm)	P *
Circle fitting constraint with BP neural network	2.2	3.0	−5.8	9	2.4	2.7	−4.2	8	1.6	2.1	−5.7	9
Axis align bounding box with BP neural network	2.7	3.1	5.9	5	3.7	4.7	9.8	2	2.2	2.6	4.2	7
Circle fitting constraint with polynomial curve fitting	5.9	7.4	−13.8	4	4.5	5.8	−12.0	10	15.8	23.9	−59.6	12
Axis align bounding box with polynomial curve fitting	7.5	8.8	−21.6	4	4.7	5.7	10.6	5	16.1	24.1	−64.5	12

* P stands for the location of the specific point where the Maximum Error occurs.

## Data Availability

The raw data supporting the conclusions of this article will be made available by the authors on request.

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
