# Peer review of "Automated Recognition and Measurement of Corrugated Pipes for Precast Box Girder Based on RGB-D Camera and Deep Learning"

_sensors, 2025, doi:10.3390/s25092641_

Round 1
Reviewer 1 Report
Comments and Suggestions for Authors
The paper "Automated recognition and measurement of corrugated pipes for precast box-girder based on RGB-D camera and deep learning" proposes a method for corrugated pipe detection. The authors' work focuses on building point cloud datasets using RGB-D camera and extracting pipeline point clouds using deep learning methods. While the results appear promising, several points need to be addressed:
1. The authors should clearly articulate the innovative aspects of this paper in relation to existing literature in the field in the introduction section.
2. The point cloud images presented in the paper are too small in size. A clear, detailed local point cloud image should be provided as a reference to facilitate accurate evaluation of the authors' work.
3. What are the specific procedures for color-assisted registration? More details and information are needed.
4. The work primarily utilizes existing methods. The innovation is limited, and the contribution is insufficient.
5. While the authors investigated the sensitivity of color features in registration, why wasn't there further exploration of the effectiveness of RGB features versus depth features in the point cloud semantic segmentation task?
Author Response
We sincerely appreciate the constructive and insightful feedback on our manuscript.
In response to your suggestions, we have made extensive corrections to our previous draft.
Please see the attachment.

Reviewer 2 Report
Comments and Suggestions for Authors
This paper proposes a method that utilizes an RGB-D camera and deep learning. The content is within the scope of the Journal. Below are comments aim to clarify technical descriptions and refine the presentation for better readability and impact.
- The Introduction discusses various traditional and non-contact detection technologies but does not clearly articulate how the proposed method improves or integrates these existing technologies. It would be beneficial to include a direct comparison or a detailed discussion on the incremental benefits of the proposed method over existing ones.
- In Section 2 on Methodology, while the sequence of steps is well-outlined, the explanation of each step could be improved by specifying how each contributes to the accuracy and efficiency of the overall system. For instance, how does the choice of segmentation algorithm specifically impact the performance in noisy or cluttered environments?
- The Introduction could better set the stage for the need for automated recognition and measurement of corrugated pipes by discussing the economic or safety impacts of errors in current manual measurement techniques more explicitly.
- In the Second paragraph of Introduction Section, it is suggested to have few sentences to review the applications of non-contact method in civil engineering applications, especially in structural health monitoring, and then narrow to the application to locate pipes. For example, the application in structural crack detection https://doi.org/10.1016/j.engappai.2022.105478 and structural displacement measurement 10.55092/sc20240003 as well as the potential for practical application https://doi.org/10.1016/j.engstruct.2024.118809.
- The paper introduces several advanced technological concepts and methodologies but does not consistently explain how each directly impacts the project's goals. For instance, the link between semantic segmentation and its role in enhancing measurement accuracy could be more explicitly detailed.
- In the Introduction, there is a missing space in the citation "[16 – 20]". It should be corrected to "[16–20]" for consistency with other citation styles in the document.
- In Section 3.1, the sentence "The camera is capable of outputting both RGB images and dense point cloud data, with each point cloud data containing XYZ (spatial coordinates), RGB (color), and SNR (signal-to-noise ratio) information." contains a redundancy. Consider revising to "The camera can output both RGB images and dense point cloud data, each containing XYZ (spatial coordinates), RGB (color), and SNR (signal-to-noise ratio) information.
- In the Methodology section, the term "pipes data" appears frequently. It should be revised to "pipe data" for grammatical accuracy.
Author Response

(The authors gave the same response as above.)
